META-RESEARCH

# Task specialization across research careers

**Abstract** Research careers are typically envisioned as a single path in which a scientist starts as a member of a team working under the guidance of one or more experienced scientists and, if they are successful, ends with the individual leading their own research group and training future generations of scientists. Here we study the author contribution statements of published research papers in order to explore possible biases and disparities in career trajectories in science. We used Bayesian networks to train a prediction model based on a dataset of 70,694 publications from PLoS journals, which included 347,136 distinct authors and their associated contribution statements. This model was used to predict the contributions of 222,925 authors in 6,236,239 publications, and to apply a robust archetypal analysis to profile scientists across four career stages: junior, early-career, mid-career and late-career. All three of the archetypes we found - leader, specialized, and supporting - were encountered for early-career and mid-career researchers. Junior researchers displayed only two archetypes (specialized, and supporting), as did late-career researchers (leader and supporting). Scientists assigned to the leader and specialized archetypes tended to have longer careers than those assigned to the supporting archetype. We also observed consistent gender bias at all stages: the majority of male scientists belonged to the leader archetype, while the larger proportion of women belonged to the specialized archetype, especially for early-career and mid-career researchers.

**NICOLAS ROBINSON-GARCIA\*, RODRIGO COSTAS, CASSIDY R SUGIMOTO, VINCENT LARIVIÈRE AND GABRIELA F NANE**

**\*For correspondence:** elrobinster@gmail.com

## Introduction

Successful research careers are built on concepts such as leadership (*Shen and Barabási, 2014*), productivity (*McKiernan et al., 2019*; *Reskin, 1979*), and impact (*Radicchi et al., 2009*; *Petersen et al., 2014*). But evidence suggests that the design of a unique career path built on individualistic success may hamper the way in which science is actually produced (*Milojević et al., 2018*). Collaboration has become essential and ubiquitous (*Guimerà et al., 2005*; *Mongeon et al., 2017*); however, the increase in team size may come at a cost for those who are not in leading roles (*Milojević et al., 2018*). The overreliance on past success in terms of accrued credit through publications and citations (*Merton, 1968*) may both reduce the scientific careers of team players and introduce gender biases (*Cole and Zuckerman, 1984*; *Macaluso et al., 2016*; *Larivière et al., 2013*), discouraging women to

pursue careers in academia (*Gaule and Piacentini, 2018*; *Huang et al., 2019*). The heterogeneity in scientists' profiles realizes the need for distribution of labor (*Larivière et al., 2016*). However, there is still a lack of understanding of how research profiles differ from each other, and how they are associated with career stages (*Laudel and Gläser, 2008*).

The goal of this study is to analyze the relation between task specialization and career length of scientists. Do specific profiles of scientists have shorter research careers than others? How do profiles relate to gender? Are these differences also reflected in productivity and citations? To answer those questions, we develop a Bayesian network-that is, a probabilistic graphical model-to predict the specific contributions scientists made to each of their publications throughout their career. We then profile researchers based on contribution statements and explore how those profiles evolve

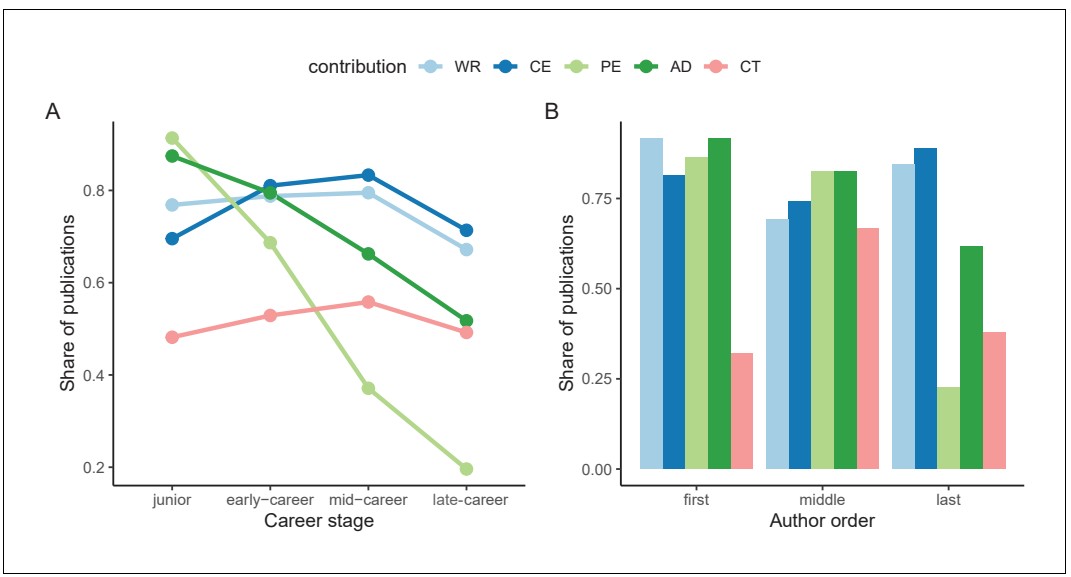

**Figure 1.** Distribution of contributions by career stage and author order. (**A**) Share of publications of authors by contributorship at each career stage. (**B**) Share of publications of authors by contributorship based on their author position in each paper. Only publications with at least 3 authors are included for B. Career stages: junior stage (< 5 years since first publication); early-career stage ($\geq$ 5 and < 15 since first publication); mid-career stage ($\geq$ 15 and < 30 years since first publication); and full career stage ($\geq$ 30 years since first publication). WR (wrote the paper); AD (analyzed the data); CE (conceived and designed the experiments); CT (contributed reagents/materials/ analysis tools); PE (performed the experiments); NC (number of contributions).

throughout their careers. We investigate how profiles at each career stage affect career length, with a particular focus on the relationship with the perceived gender of the scientist. Finally, we examine the relationship between profiles and bibliometric characteristics, such as research production and scientific impact.

Our seed dataset contains a total of 70,694 papers authored by 347,136 scientists from PLoS journals in the Medical and Life Sciences fields. Author names are disambiguated using a rule-based scoring algorithm (*Caron and van Eck, 2014*). Each author has also been linked to their bibliometric data from Web of Science. We restrict our dataset to the Medical and Life Sciences to make it more homogeneous and avoid disciplinary differences in task distribution. We assign papers to fields by identifying the journal to which each of the references of the publications in our dataset belong. We then assign to each publication the field from which most of its references come. Finally, we only include those which are assigned to the Medical and Life Sciences fields. Further details are provided in the Materials and methods section.

We then build a probabilistic model to predict authors' contribution to publications, based on a set of bibliometric variables. This model allows us to extend our analysis from the initial

dataset to the complete publication history of these authors. We reconstruct the publication history of 222,925 authors from our original dataset and predict, for each author, the probability of conducting a given contribution on each of their publications. Based on the new dataset of predicted probabilities of contributorship, we divide scientists' careers into four stages and conduct an robust archetypal analysis (*Eugster and Leisch, 2011*) by stage. This allows us to identify differences in scientific profiles by stage and gender, and explore differences in scientific paths.

## Results

### Contribution statements and predicting variables

Five types of contribution statements are identified in the contribution dataset: wrote the paper (WR), conceived and designed the experiments (CE), performed the experiments (PE), analyzed the data (AD), and contributed reagents/materials/analysis tools (CT). The number of contributions (NC), that is, the sum by paper of the contributorships each author reports, is also considered. These contributions are assumed to be related with author order (*Milojević et al., 2018*; *Mongeon et al., 2017*; *Sauermann and*

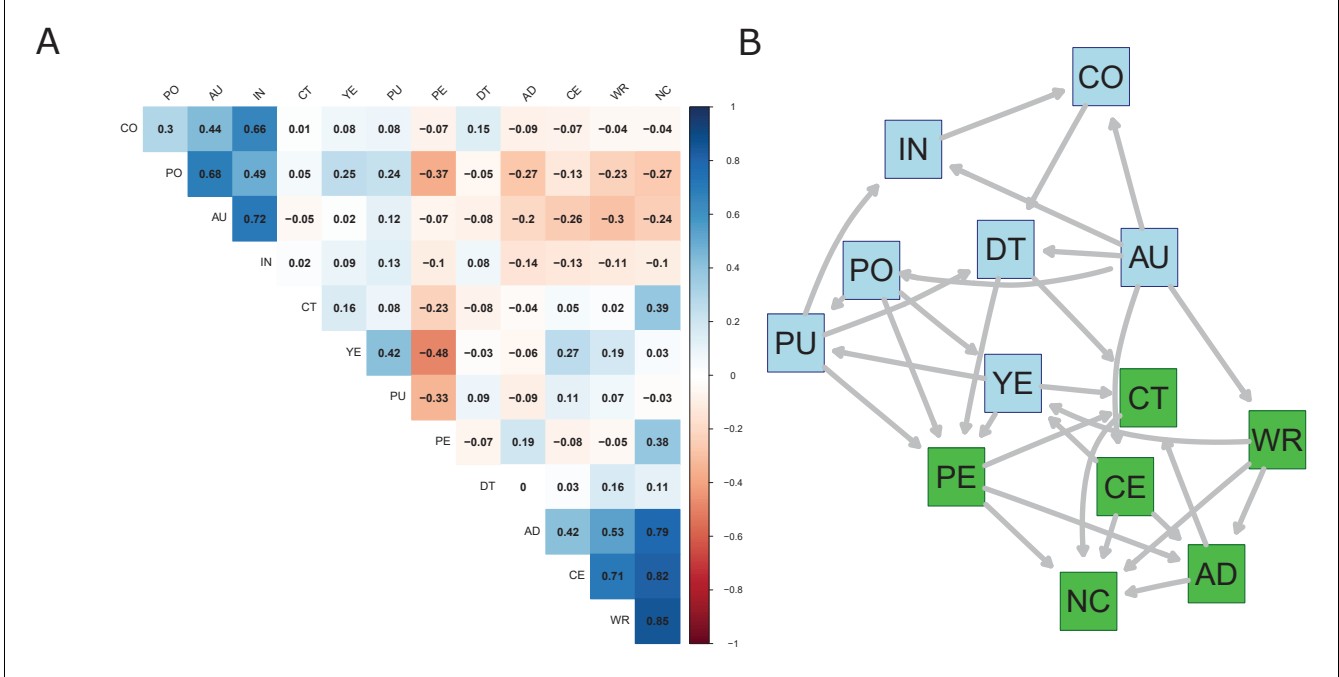

**Figure 2.** Mixed correlation matrix of contributorship and bibliometric variables (A) and the Bayesian network used for predicting contributorship (B). Contribution variables are in green, bibliometric variables are in blue. *Bibliometric variables*: PO (author's position); AU (number of authors); DT (document type); CO (number of countries); IN (number of institutions); YE (years since first publication); PU (average number of publications). *Contribution variables*: WR (wrote the paper); AD (analyzed the data); CE (conceived and designed the experiments); CT (contributed reagents/materials/analysis tools); PE (performed the experiments); NC (number of contributions).

The online version of this article includes the following figure supplement(s) for figure 2:

**Figure supplement 1.** Bayesian network structure used for predicting contributorship highlighting whitelisted arc relations.

*Haeussler, 2017*), with first and last positions in author order reflecting leadership (*Chinchilla-Rodríguez et al., 2019*), as per the recommendations of the *International Committee of Medical Journal Editors, 2015*. *Figure 1* relates career stage and author order with contribution role. We define four career stages: junior (< 5 years since first publication), early-career (≥ 5 and < 15 years since first publication), mid-career (≥ 15 and < 30 years since first publication) and late-career (≥ 30 years since first publication). These four stages are defined in consistency with other classifications of career stages in the literature (*Laudel and Gläser, 2008*; *Milojević et al., 2018*; *European Commission, 2016*).

The distribution of reported contribution roles by career stage shows that earlier stages are more often associated with performing experiments and analyzing data, and that this contribution decreases as individuals become more senior. Writing the manuscript and contributing reagents and tools increase over time, with a decline in the late-career stage. Conceiving and designing the experiments demonstrates a

modal shape, where early-career and mid-career stages are the ones in which these tasks are more prominent. In terms of labor distribution, first authors are heavily associated with all contributions, with the exception of contributing tools, reagents, data, and other materials. Middle authors report to be less involved in writing tasks or in the design and conception of experiments but are associated with contributing resources to a much greater extent. Last authors report contributing mostly to the design and conception of experiments as well as to writing tasks, and to a lesser extent to the performance of experiments.

Bibliometric indicators are employed as predictors of contributorship. Two types of bibliometric variables are included: paper-level and author-level. Paper-level variables are document type (DT), number of authors (AU), number of countries (CO), and institutions (IN) to which authors of the paper are affiliated. Author-level variables include their position in the authors' list (PO), number of years since they published their first publication (YE) and the average number of publications per year (PU).

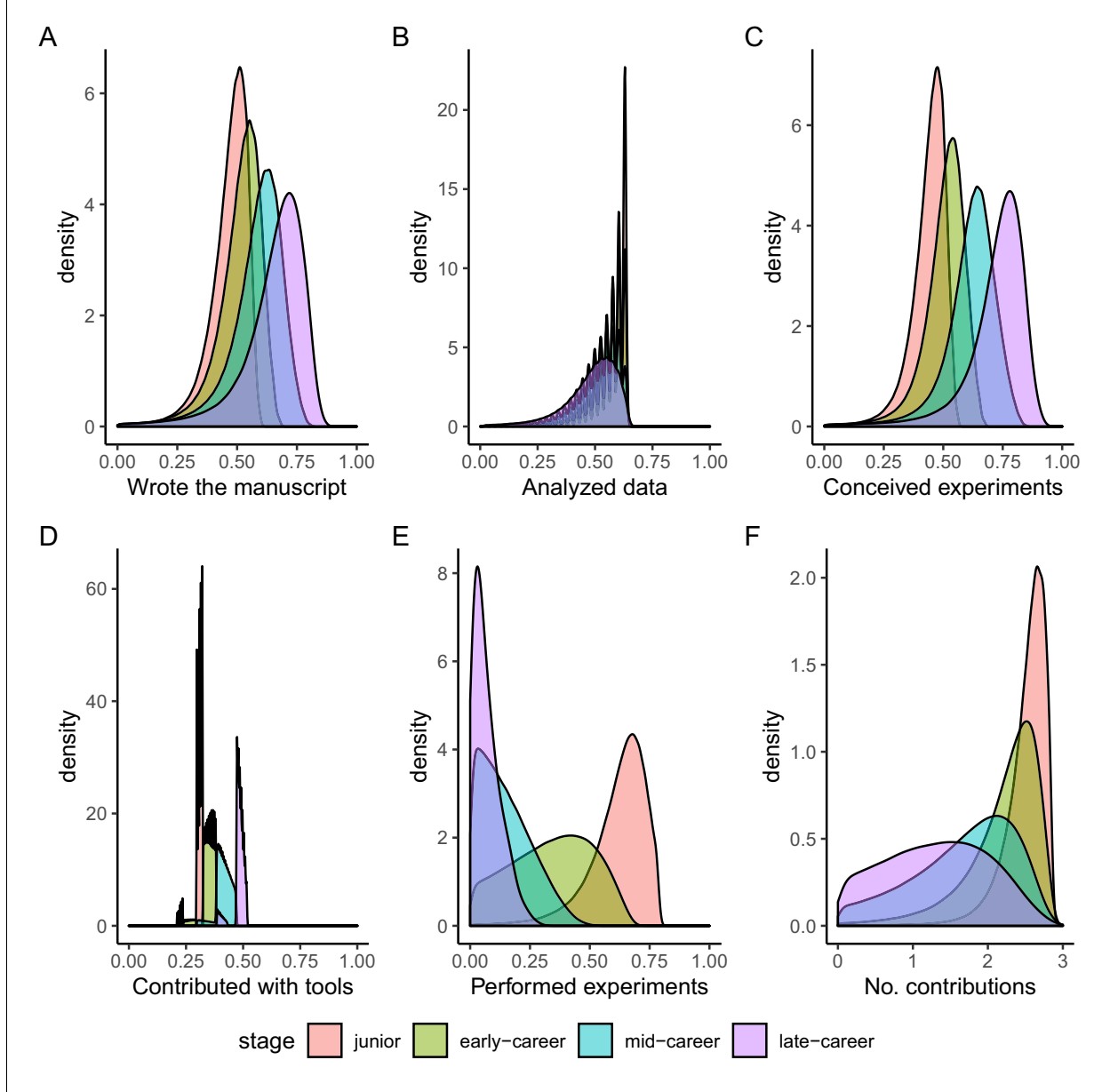

**Figure 3.** Probability density functions of contribution roles predicted using the Bayesian Network model. Distributions are aggregated by career stage. (A) Probability distributions for the contributorship *Wrote the manuscript.* (B) Probability distributions for the contributorship *Analyzed the data.* (C) Probability distributions for the contributorship *Conceived and designed the experiments.* (D) Probability distributions for the contributorship *Contributed with tools.* (E) Probability distributions for the contributorship *Performed the experiments.* (F) Probability distributions for estimated *Number of contributions* of an author. Red color refers to scientists' junior stage, green to early-career stage, blue to mid-career stage and purple to late-career stage.

Figure 2A depicts the coefficients of a mixed correlation matrix of the contributorship and bibliometric data, while Figure 2B illustrates the Bayesian network used for predicting the contribution of a researcher for a given publication. The highest correlations within types of contributorship are between writing the manuscript and conceiving and designing the experiments (0.71), while the rest of contributorship variables exhibit low correlations. In the case of bibliometric variables, there is a moderate positive correlation between number of countries and institutions (0.66), author position and number of authors (0.68), and number of authors and number of institutions (0.72). A strong positive monotone relation between the number of

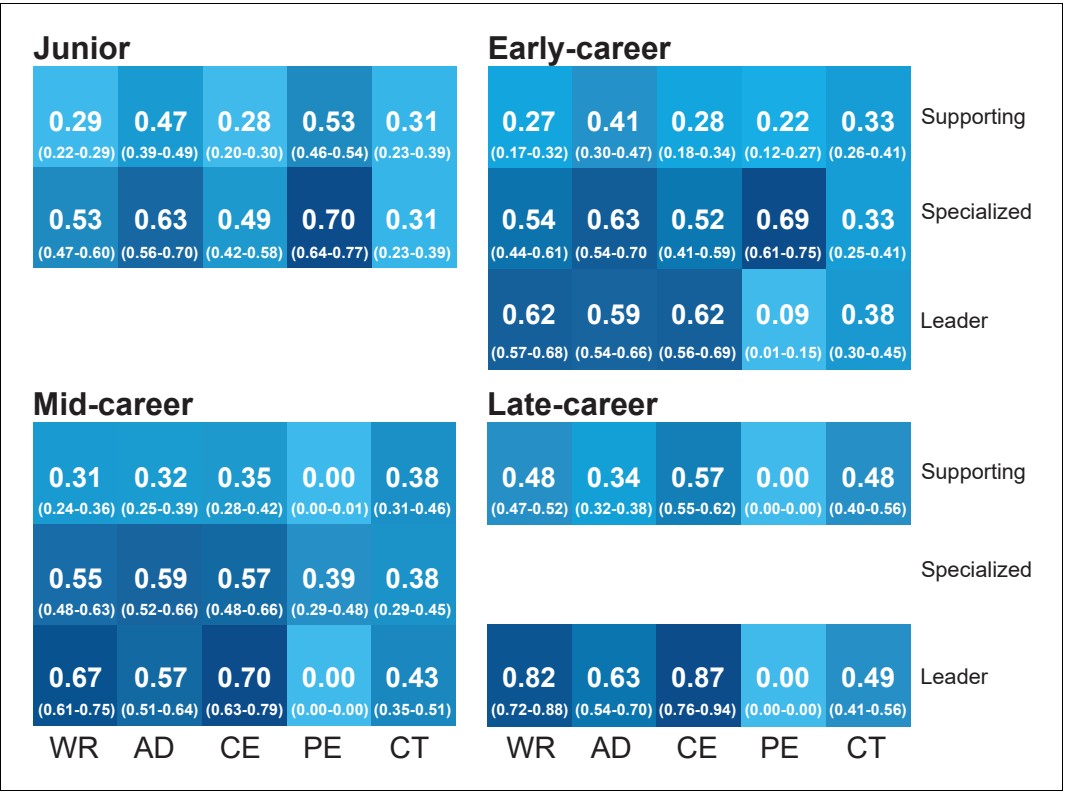

**Figure 4.** Coefficient values of contributorships by archetype, per career stage. Two archetypes are identified in the junior stage (*Specialized* and *Supporting*), three have been identified for the early- and mid-career (*Leader*, *Specialized* and *Supporting*) and two have been identified for the late-career stage (*Leader* and *Supporting*). Uncertainty intervals of coefficients are shown in brackets. Color grades reflect the value of the parameters. Contributions statements: WR, wrote the manuscript; AD, analyzed data; CE, conceived and designed the experiments; PE, performed the experiments; CT, contributed with tools.

The online version of this article includes the following figure supplement(s) for figure 4:

**Figure supplement 1.** Screeplots of the residual sum squares (RSS) which allows determining the number of archetypes for each career stage.

contributions and either writing the manuscript (0.85), conceiving the experiments (0.82) or analyzing the data (0.79) is observed. The number of contributions seem therefore to be associated with those type of contributions. Weak monotone negative relationships are suggested by correlations between the number of contributorships and bibliometric variables. Negative correlations are observed between performing the experiments and position in authors list, years since publication and average number of publications. Weak to moderate negative correlations are observed between contributorship variables and the number of countries and institutions, author's position, and number of authors of a publication.

*Bayesian network model for predicting contributorship*

We model our dataset using a Bayesian network (BN) to be able to predict contribution roles of scientists for their publications based on the bibliometric information of the given publications. The aim here is to expand our original dataset to the complete publication history of the 347,136 researchers from the Medical and Life Sciences who had published at least one paper in our PLOS seed dataset. A BN is a probabilistic graphical tool used to model multivariate data (*Nielsen and Jensen, 2009*). The variables are denoted as nodes in the network, whereas the arcs denote influences between variables, typically quantified as dependencies. BN accounts not only for dependencies between the predictor variables and variables of interest, but also for dependencies between predictor variables.

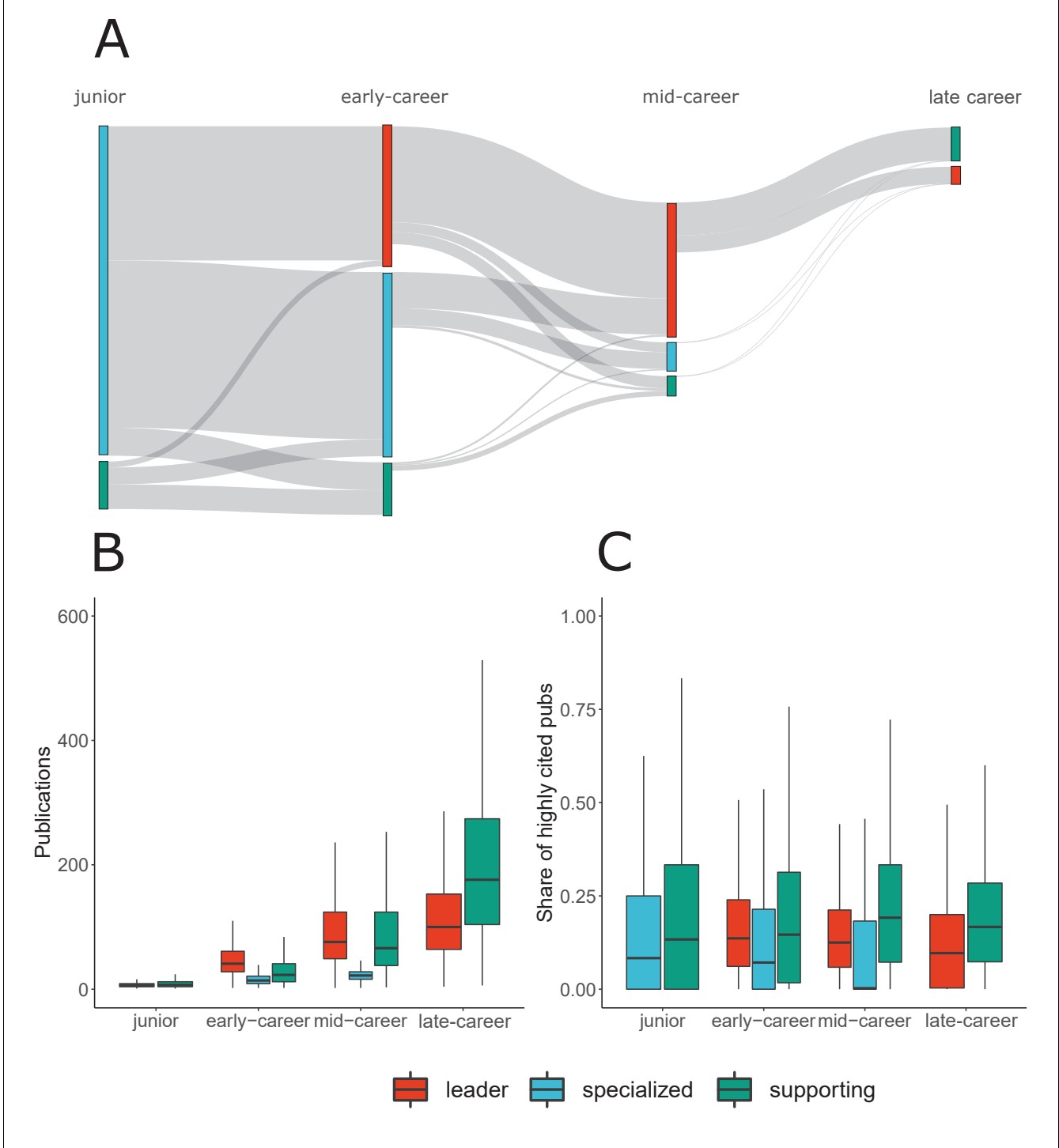

**Figure 5.** Career trajectories, productivity and citation impact boxplots by archetype. (A) Sankey diagrams indicating the number of scientists by archetype at each career stage and transitions from one stage to the next, including changes on researchers' archetype. (B) Productivity boxplots, by archetype and career stage. This is calculated based on the cumulative number of publications scientists had authored at each given stage. (C) Share of highly cited publications boxplots by archetype and career stage. Highly cited publications are defined as those which are among the 10% most highly cited publications in their field and year of publication. Red refers to the Leader archetype, Blue refers to the Specialized archetype and Green refers to the Supporting archetype.

The online version of this article includes the following figure supplement(s) for figure 5:

**Figure supplement 1.** Effect size for the differences between archetypes within each career stage for A number of publications and B share of highly cited papers.

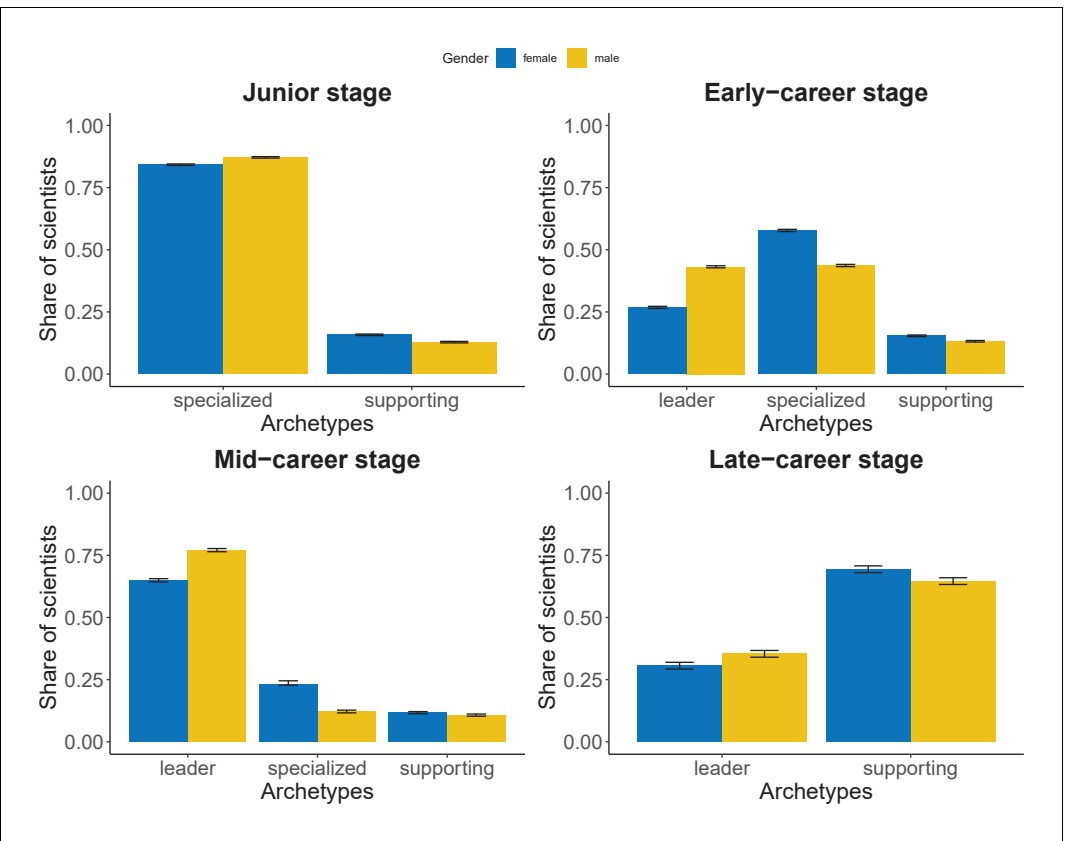

**Figure 6.** Estimated proportion of scientists, along with 95% confidence intervals, by gender and career stage for each archetype. Top-left panel refers to the junior stage in which only two archetypes are present: specialized and supporting. Top-right refers to the early-career stage. Bottom-left refers to the mid-career stage. Bottom-right refers to the late-career stage, again here only two archetypes are observed: leader and supporting. Blue refers to women scientists and yellow to men scientists.

The online version of this article includes the following figure supplement(s) for figure 6:

**Figure supplement 1.** Sankey diagram indicating the number of male scientists by archetype at each career stage and transitions from one stage to the next, including changes on researchers' archetypes.

**Figure supplement 2.** Sankey diagram indicating the number of female scientists by archetype at each career stage and transitions from one stage to the next, including changes on researchers' archetypes.

**Figure supplement 3.** Effect sizes for proportion tests to identify differences by gender and archetype at each career stage.

This characteristic, along with the forthright graphical representation, makes BNs an attractive choice to model dependent multivariate data.

*Figure 2B* shows the structure of the obtained BN. Five types of contributions along with the number of contributions (in green) of scientists are predicted using the seven bibliometric variables (in blue). The structure of the BN has been obtained by using a hybrid data-learning algorithm called Max-Min Hill Climbing (MMHC) (*Tsamardinos et al., 2006*), along with the constraint that bibliometric variables are influencing contributorship variables. That is, if an arc between bibliometric and contributorship variables is present in the structure, then it should be directed to the contributorship variable. Furthermore, the structure of the network has been tested for robustness. The strength of the arcs, i.e., relationships between variables, has been investigated using the bootstrap procedure, with 50 repetitions. Only the arcs that were present in 80% of the repetitions have been considered and are depicted in *Figure 2B*.

We evaluate the predictive power of the obtained BN using k-fold cross-validation. That is, the data has been repeatedly divided in 10 random folds, of which 9 have been used to learn the BN structure using the MMHC algorithm together with the aforementioned

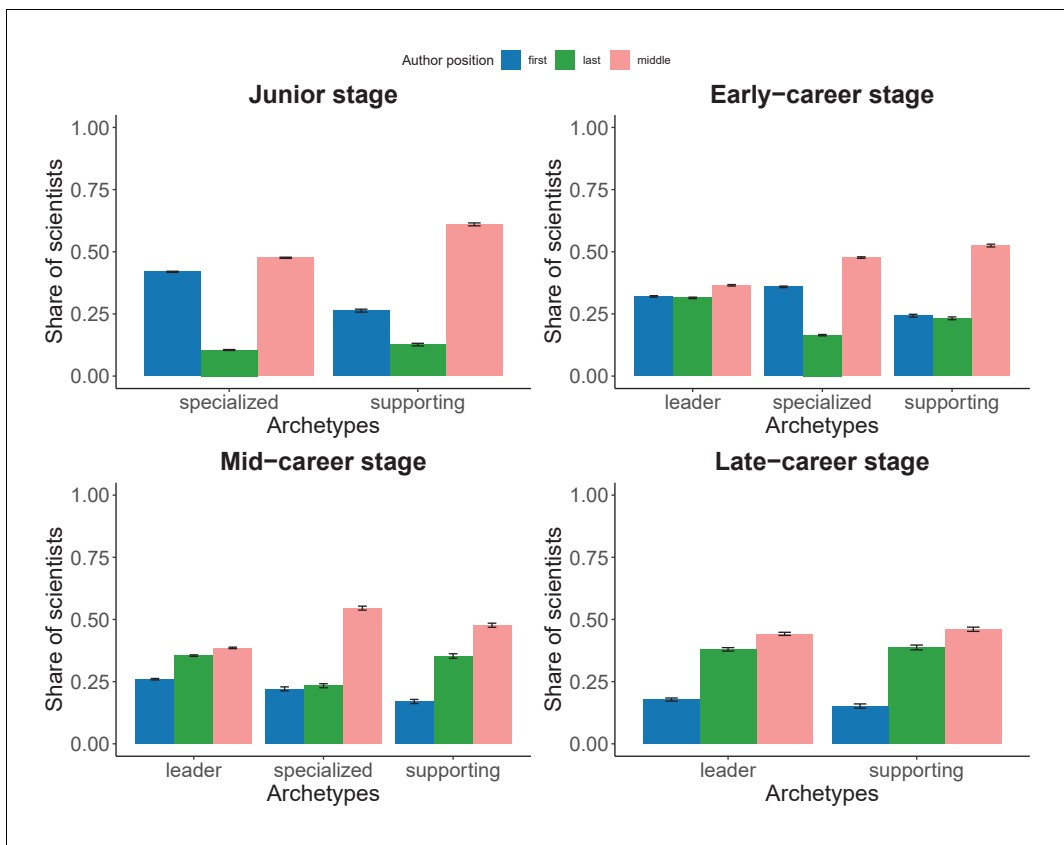

**Figure 7.** Percentage of scientists by author position, along with 95% confidence intervals, for each archetype and career stage. Top-left panel refers to the junior stage in which only two archetypes are present: specialized and supporting. Top-right refers to the early-career stage. Bottom-left refers to the mid-career stage. Bottom-right refers to the late-career stage, again here only two archetypes are observed: leader and supporting. Blue refers to share of scientists publishing as first authors, green refers to those publishing as middle authors, and pink refers to those publishing as last authors.

The online version of this article includes the following figure supplement(s) for figure 7:

**Figure supplement 1.** Effect sizes for differences in proportions by author position and archetype at each stage.

constraints. The contributions were then predicted for the remaining fold. The procedure has been repeated for each of the 10 folds and results on the prediction errors reported in the Materials and methods section. The predictive performance of the BN has been shown to be extremely good, with an average classification error rate of between 6-8% for all contributorships and a mean squared error (MSE) of 0.12 for the predicted NC. The BN is used to predict the contributions for the complete publication history of a subset of 222,925 scientists who have published in PLOS journals, for a total of 6,236,239 publications. Each contribution is predicted as the probability that an author has performed a given contribution on a publication. We further investigate the distributions of the predicted contributorships.

When distinguishing by career stage (*Figure 3*), the densities clearly depict differences in contributorships. Performing the experiments is the most discriminative contributorship type, with junior scientists more likely be associated with this contribution. The more scientists advance in their career, the less likely that they will perform the experiments. Albeit less dramatic, the same discriminative pattern can be observed for analyzing the data and for the total number of contributorships, with decreasing association by age. Inversely, the contribution roles of wrote the manuscript, conceived the experiments, and contributed with tools are more likely for advanced career stages.

## Profiling scientists using robust archetypal analysis

We aggregate the predicted contributorships at the individual level and by career stage to profile scientists based on their contributorship patterns. To avoid the effect of contributorship outliers, we aggregate researchers' contributorships by choosing the median predicted contributorship of publications for each career stage. We perform a robust archetypal analysis (RAA) to identify types of scientists based on their contributorships (*Eugster and Leisch, 2011*). Archetypes accentuate distinct features of scientists based on contribution data. Robust archetypal analysis identifies ''prototypical types'' of the multivariate aggregated contributorship dataset, correcting for outlier effects in the data. Each of these ''prototypical types'' or archetypes is represented as a convex combination of researchers in the aggregated contributorship dataset and, in turn, each researcher is well described by a convex combination of these archetypes.

We consider archetypes of scientists at each career stage. A residual sum of squares (RSS) analysis for different archetypes reveals that using two archetypes for the junior and late-career stages, and three for early-career and mid-career stages results in significantly smaller RSS. *Figure 4—figure supplement 1* reveals the screeplots of RSS per career stage, where the elbow criterion supports the choice of number of archetypes per career stage. The influence of contributorships within each archetype is captured by corresponding coefficient values. Coefficients of each archetype (Leading, Specialized and Supporting) per career stage are presented in *Figure 4*. Low values indicate low prevalence of corresponding type of contributorship, whereas high values indicate a high contribution to the archetype.

A first notable observation is that differences in contributions are remarkably small for certain archetypes throughout career stages. Given that the archetypes at each stage have common characteristics, we maintain the same profile naming across stages. Three archetypes are identified. The Leader is characterized by high coefficient values for all contributions, except for PE, indicating a high prevalence of each contribution role, and especially on WR and CE. The Specialized archetype is characterized by high coefficient values for PE and AD. A trend analysis for this archetype indicates a shift between PE and AD contributions. The third archetype is referred to as the Supporting, and is characterized by generally low values for all contributorships. This is the least discriminatory archetype.

At the junior stage, we observe two archetypes: Specialized and Supporting. Both are characterized by scientists reporting more than two contributions per paper. For the Specialized archetype, the most prevalent roles are on PE and AD, although they show higher coefficients than Supporting for all contributorships except CT (with a marginal difference). At the early-career stage, three archetypes are obtained, with a clear difference on PE between Leader and Specialized. These three archetypes are maintained during the mid-career stage, with the most notable difference being the shift between AD and PE for the Specialized, that now exhibits a higher probability of conducting the former than the latter. In the late-career stage, the Specialized archetype is no longer identified, and again two archetypes emerge. Both archetypes show low probabilities on PE, while the Leader is characterized by a higher probability on WR and CE. Overall, RAA shows that BN's predictions can accurately capture the diversity of archetypes of scientists and are sufficiently discriminating.

Uncertainty of the coefficient values has been accounted for to illustrate the robustness of the obtained archetypes, per career stage. The uncertainty intervals display small variations around the initial coefficients, which confirms the robustness of the archetypes. The large differences as well as similarities in contributions are well preserved by the uncertainty intervals.

## Career paths, productivity and citation impact

Similarities between the archetypes are identified at each career stage, demonstrating the stability of the classification by scientific age (*Figure 4*). In turn, each scientist can be represented as a weighted combination of the archetypes. For a given scientist, the weights, or $\alpha$ scores, corresponding to each archetype determine the researchers' assignment to one of the two or three archetypes. Here, we assign researchers to archetypes based on the highest weight. The assignment can be done for each career stage, which naturally leads to a career path.

*Figure 5A* presents the assignment of researchers to the archetypes and their evolution over the four career stages, using the maximum coefficients and the median aggregation method. However, we observe some patterns by

archetype. Out of the 222,295 scientists included in the dataset, 27,714 reached the late-career stage. We observe that there is little attrition, regardless of the archetype to which scientists belong, between the junior and early-career stage (93% for junior Specialized and 83% for Supporting authors). At the early-career stage, when the Leader archetype emerges, the advantage of those exhibiting a Leader profile becomes evident: 84% of scientists who belong to the Leader archetype in their early-career reach the next career stage, while 30% and 16% of Specialized and Supporting scientists progress to mid-career stage respectively. The cost is even higher from mid-career to late-career, with 37% of Leader profile scientists, and only 1% and 2% of Specialized and Supporting authors reaching the last career stage.

Furthermore, 98% of scientists reaching the late-career stage exhibited a Specialized archetype in their junior stage, and 67% of those reaching this last career stage have consistently displayed a Leader profile in early- and mid-career stages. Shifts across archetypes appear more likely at earlier career stages, as well as from the Leader archetype to the other two archetypes (but not vice versa). Even though most of the scientists reaching the late-career stage belong to the Leader archetype in their mid-career stage, 66% of late-career researchers are in a Supporting role, although they remain involved in more than one contributorship type.

When comparing archetypes by number of publications (*Figure 5B*), we observe almost no differences in publication rates in the junior stage. Nonetheless, differences emerge for later career stages. Except for the late-career stage, where Supporting scientists are the most productive, the Leader archetype exhibits higher productivity, followed by Supporting. Specialized scientists appear to be much less productive than scientists assigned to the other two archetypes in the early- and mid-career stages. This pattern is also observed for Specialized, in the case of citation impact. However, differences in terms of share of highly cited publications between the Leader and the Supporting archetypes are much smaller, with the latter exhibiting higher values.

We investigated whether the differences are statistically significant using Wilcoxon rank sum test (*Wilcoxon, 1945*). All group comparisons between archetypes within each career stage reveal statistically significant differences. Furthermore, effect sizes to evaluate the strength of the differences are reported in *Figure 5—figure*

supplement 1, along with their confidence intervals. We observe large effect sizes on the differences in productivity for early- and mid-career stages, a medium effect size for late-career and a small effect-size for junior stage. Despite the low p-values (all below $2.2E-16$) and the apparent difference in median share of highly cited publications between the specialized archetype and the other two archetypes of the mid-career stage, we observe that the effect sizes are small across all career stages.

### Archetypes and gender

*Figure 6* shows that scientists are unevenly distributed by gender in each archetype. Note that scientists from different generations are included in the analysis, therefore, caution should be expressed in drawing any conclusion on the number of scientists by gender that reach the late-career stage. The share of women who reach the late-career stage is affected by the generational diversity of scientists and hence we make comparisons only within career stage. We observe a gender disparity especially in the early- and mid- career stages. The share of men is higher for the Specialized archetype at the junior stage, and for the Leader archetype at the early- and mid-career stages. The second most frequent is the Specialized archetype, with few men in the Supporting archetype, except for the late-career stage. Women are less likely to appear as the Leader archetype in the early- and mid-career stages. Whereas 87% of men in the junior stage have a Specialized archetype, 43% and 77% in the early- and mid-stage are designated as Leaders; 84% of women in junior are Specialized, and only 27% and 65% in early- and mid-career stages show a leading profile. The gender distribution becomes more balanced again at the late-career stage, where 35% of men and 31% of women are in the Leader archetype. In summary, women appear to group within the Specialized archetype in the early-career stage, and show similar distributions to that of men at the other career stages, although the shares of the Leader archetype are consistently lower to that of men. These differences on the distribution of scientists by archetype and gender and how they might affect their trajectory is made more evident in *Figure 6—figure supplement 1* and *2*.

We employed two-proportion z tests, based on Pearson's chi-squared test statistic, to investigate whether the differences in proportions within career stage are statistically significant. A 95% confidence interval of the differences in

proportion allowed us to compute confidence intervals around the estimated percentages. As expected, the confidence intervals along with the very small p-values (all below $2.2E - 16$) indicate that the differences are statistically significant, hence it is unlikely that the observed differences to have occurred by chance. Effect sizes have been computed using *Cohen, 2013*. We observe a medium effect size on differences by gender for Leader and Specialized profiles at the early- and mid-career stages (*Figure 6—figure supplement 3*).

### Archetypes and author position

We analyze the relationship between author order and archetypes by career stage. *Figure 7* shows the share of papers by archetype and career stage of scientists based on their author position. Middle authorships occupy a larger share of publications irrespective of the archetype or career stage, which is a consequence of the fact that any paper with more than three authors, most authors are in middle positions. We do observe, however, variation in middle authorship by career stage. At the junior stage, middle authorships account for half of the papers from Specialized scientists, while Supporting scientists occupy a middle position in almost 75% of their publications. In the early-career stage, the Leader archetype emerges, exhibiting a more balanced share of publications between first (32%), middle (37%) and last positions (32%). Specialized scientists publish a slightly higher share as first authors (36%) but almost in half of their papers appear in middle positions (48%). The Supporting archetype publishes more than half of their papers as middle authors (53%), evenly distributed between first and last authored publications.

At the mid-career stage, Leader scientists start to shift to last positions (36%), with only 26% of their publications being first authored. Specialized scientists become the middle authors in 55% of their publications and are last authors on 23% of their publications. Supporting scientists, however, position themselves as last authors in 35% of their publications. The Specialized archetype disappears in the late-career stage. The Leader and Supporting archetypes show similar distributions of publications according to their author position, revealing that at this stage, author position is more related with seniority than contributorship.

Similar to the gender analysis, we have evaluated whether the differences in proportions are statistically significant. All pairwise tests reveal

statistical significance, as supported also by the very small confidence intervals. The effect sizes are reported in *Figure 7—figure supplement 1*. We observe a large effect size for specialized authors in the junior stage and supporting authors in the late-career between being first or last author. Large effects sizes are also observed between being first or middle author for supporting authors across all stages, as well as specialized at their mid-career stage and leaders and their late-career stage. Finally, we observe a large effect size between having a middle or last position for specialized and supporting authors at the junior and early-career stage, and for specialized authors at the mid-career stage. For the rest, we report between middle and small effect sizes.

## Discussion

The assessment of researchers has been under scrutiny for some time (*McKiernan et al., 2019*; *Moher et al., 2018*; *Way et al., 2019*; *Weingart, 2005*). They are immersed in a reward system that evaluates them individually following uniform expectations of leadership and excellence (*Bol et al., 2018*; *Merton, 1968*; *Reskin, 1977*). Recent evidence shows an increasing need for a larger and more stratified scientific workforce (*Milojević, 2014*; *Larivière et al., 2016*; *Newman, 2004*; *Wuchty et al., 2007*) which necessarily involves a reconceptualization of research careers and considering a breadth of profiles for which specific paths should be considered. Larger teams require a distribution of tasks which will translate on individuals specializing on certain roles (*Larivière et al., 2016*). Here we identify and characterize such diversity of profiles by career stage, by combining contribution statements with bibliometric variables and applying a machine learning algorithm to predict contributions.

We find that scientists exhibit different archetypes at different stages, following many paths during their career trajectory. Some paths, however, come at a cost. Out of the 222,295 scientists included in our dataset, only 12% reached the late-career stage. While this should not be striking, as scientists of different ages are included in our analysis, it is worth noting that the vast majority (98%) of these scientists displayed a Specialized archetype in their junior stage. Even though most of them belonged to the Leader archetype in their early- and mid-career stages, scientists at the late-career stage

mostly exhibit a Supporting archetype (66%). This could be happening because many scientists adopt a secondary role when they reach seniority, leaving the leading role to their younger colleagues.

The names assigned to each archetype are figurative but reflect an implicit hierarchy in science. This hierarchy exists at each career stage, indicating that the diversity of profiles is not the result of scientists evolving in their career trajectory and adopting different roles, but that diverse archetypes exist between and within career stages. The robust archetypal analysis identified no Leader archetype at the junior stage, when scientists are still 'earning their stripes', nor are there Specialized scientists in the late-career stage. Such reality enters into conflict with the current expectations of research careers, which consider roles to be attached with career stages and steps that must be made to progress. Our findings have important policy implications as they indicate that scientists' career design may be at odds with the way science is produced, and suggest a complete reform wherein reproduction of Leaders is not the only model of success (*Milojević et al., 2018*).

Our results demonstrate the high versatility of the Leader archetype: scientists with this profile are able to move seamlessly across archetypes during their careers. While there are some scientists with a Specialized or Supporting profile who manage to shift to the other three archetypes, most of the scientists fitting these archetypes in our dataset do not progress to more senior stages. Our analysis on productivity and citation impact by archetype sheds light on the mechanisms which may be affecting trajectories. Specialized scientists are less productive and have a lower share of highly cited publications than Leaders and Supporting scientists, which may serve a disadvantage for career advancement in those environments which prioritize publication productivity indicators in research assessment (*Figure 5B,C*). The lack of assessment schemes sensitive to the diversity of profiles, partly due the inappropriate use of bibliometric indicators at the individual researcher level (*McKiernan et al., 2019*; *Hicks et al., 2015*), limits the capacity of policies to correct for inequalities observed across and within archetypes. Structural changes in the academic reward system are necessary to support the advancement and retention of Specialized and Supporting scientists.

We observe consistent differences in the distribution of archetypes by gender, which may contribute to explain the higher rates of attrition for women (*Huang et al., 2019*). Early-career stage is key to the development of scientific careers, and it is at this stage that large gender differences are observed. While in the other career stages women and men exhibit a similar distribution of archetypes, women are more likely to be of the Specialized archetype in early-career, while men are more likely to be Leaders. That women disproportionately engage in technical labor–even when controlling for academic age–has been demonstrated in previous studies (*Macaluso et al., 2016*). This is consistent with general patterns in academic labor; for example, the higher service work done by women academics (*Heijstra et al., 2017*).

Contributorships are generally associated with author order (*Larivière et al., 2016*; *Sauermann and Haeussler, 2017*), based on the presumption that first and last author will have major roles, while middle authors will play a secondary role. These roles reinforce hierarchy and organizational strategies: leaders set the agenda and define lines of work, whereas technicians are prized for their ability to implement this agenda (*Latour and Woolgar, 2013*). This model, however, does not provide equal access to career advancement for all types of scientists: those showcasing a Specialized or Supporting archetype in their early- and mid-career stages have greater difficulties to progress in their research career. These obstacles affect women at a greater extent than men, as a higher proportion of female scientists adopt these roles. Our findings suggest systematic biases on the selection of individuals which may be hampering the efficiency of the scientific system to self-organize itself and assemble robust and diverse scientific teams.

## Materials and methods

The data needed to reproduce the our analysis are openly accessible at http://doi.org/10.5281/zenodo.3891055. Data were processed and extracted from a T-SQL database server held at the Centre for Science and Technology Studies-CWTS (Leiden University). Data modelling, analyses and visualization figures were conducted using the R statistical programming language version 3.6.3 (*R Development Core Team, 2020*). The Bayesian network modelling was conducted using the bnlearn package (*Scutari and Denis, 2014*). The Robust

Archetypal Analyses were conducted using the archetypes package (*Eugster and Leisch, 2009*). Visualizations were created using the ggplot2 (*Wickham, 2016*) and ggpubr packages (*Kassambara, 2020*). Mixed correlation matrix in 2A was calculated using the psych package (*Revelle et al., 2010*). Spearman rank correlations have been determined for continuous variables, tetrachoric correlations for binary random variables and biserial and polyserial correlations for mixed random variables, i.e., between binary or other discrete and continuous random variables.

Our analysis is based on two datasets: a seed dataset of contributorship statements and dataset of researchers' late-publication histories. The seed dataset combines bibliometric and contributorship data for 85,260 publications from 7 PLoS journals, during the 2006-2013 period. Although many biomedical journals have adopted contributorship statements (e.g., BMJ, The Lancet), PLoS journals provide data in an XML format which ease the data retrieval process.

This dataset is used to train a predicting model of contributorship based on bibliometric variables. The full publication histories dataset contains the complete publication history of the 222,925 authors selected from the list of publications of the first dataset. This dataset is used to predict authors' contributorship per paper and is later aggregated at the individual level to

identify archetypes of scientists per career stage. The analyses were conducted on an Intel Core i7-8550U CPU with 16GB RAM, running Microsoft Windows 10 Home Edition. The total computational time of the analyses took around 30 hours, with 20 hours being required for the data modelling.

### Contributorship statements

We used a dataset of 85,260 distinct PLoS papers published during the 2006-2013 period. This dataset was gathered from the PLoS website in combination with Web of Science data. Full account of the complete extraction procedure is provided in a previous study (*Larivière et al., 2016*). For each publication and author, a dummy value is assigned based on the tasks they performed. *Table 2* shows the list of journals together with the number of publications per journal. 88% of the publications have been published in PLoS One. Seven types of contributions were originally included in the dataset. Only five of those contributorships are being used consistently throughout the dataset. "Approved final version of the manuscript" and "Other contributions" are present in less than 5% and 20% of the papers respectively. While the former is a requirement of the ICMJE and therefore is used mostly in PLoS Medicine, the latter is not an individual category, but an aggregate containing nearly 20,000 different types of contributions. The low incidence of the

**Table 1.** Definition of variables included in the dataset.

| Acronym | Definition | Source |
|---|---|---|
| *Bibliometric variables* | | |
| PO | Author's position in the paper | WoS |
| AU | Total number of authors in the paper | WoS |
| DT | Document type. Letters are excluded | WoS |
| CO | Number of countries to which authors of the paper are affiliated | WoS |
| IN | Number of institutions to which authors of the paper are affiliated | WoS |
| YE | Number of years since first publication at the time the paper was published | WoS |
| PU | Average number of publications (full counting) per year | WoS |
| | of the author at the time the paper was published | |
| *Contribution variables* | | |
| WR | Wrote the paper | PLoS |
| AD | Analyzed the data | PLoS |
| PE | Performed the experiments | PLoS |
| CE | Conceived and designed the experiments | PLoS |
| CT | Contributed reagents/materials/analysis tools | PLoS |
| NC | Number of contributions | PLoS |

**Table 2.** Distribution of papers by journal of the seed dataset on contributions.

| Journal | No. of papers |
| --- | --- |
| PLOS ONE | 62,174 |
| PLOS GENETICS | 2408 |
| PLOS PATHOGENS | 1882 |
| PLOS COMPUTATIONAL BIOLOGY | 1684 |
| PLOS NEGLECTED TROPICAL DISEASES | 1432 |
| PLOS BIOLOGY | 697 |
| PLOS MEDICINE | 417 |

"Approved final version" contribution together with the difficulties in interpreting the "Other" contributorship led to their exclusion from the analysis.

### Bibliometric data

The bibliometric data is obtained from the CWTS (Leiden University) in-house version of the Web of Science. This database contained at the moment of analysis all publications included in the Science Citation Index Expanded, Social Science Citation Index, and Arts and Humanities Citation Index for the 1980-2017 period. Furthermore, an author name disambiguation algorithm (*Caron and van Eck, 2014*) is applied to the complete database, allowing to identify a scientist's complete publication history. This allowed us to retrieve, for each paper contained in the contribution dataset, bibliometric variables at the publication and at the author level. A set of seven bibliometric variables is considered, which is described in *Table 1* by author-publication combination. Here, we highlight the use of the variable years since first publication (YE). This variable is used to determine the age of scientists and is used later to estimate the different career stages of the individuals identified. Our use of the year of first publication as an indicator for academic age is based on previous research (*Nane et al., 2017*), in which the year of first publication is found to be the best predictor for the academic age of scientists. In the case of productivity, we use a full counting approach. While fractional counting can be considered as being more accurate from a mathematical point of view (*Waltman, 2016*), the focus here is on the previous publication experience of the author and how that might influence their role in future publications. Hence we consider full counting to suit best the purposes of the analysis.

### Merging of bibliometric and contribution data

The merging process was undertaken by matching documents by their DOI identifier and authors who had the same initials and surname in both datasets. We only included papers for which all authors were successfully matched. After this process was undertaken, we ended up with a total of 77,749 publications, containing a total of 369,537 disambiguated unique authors.

### Subject field identification

We assigned a subject field to each publication and filtered only those publications that belong to the Medical and Life Sciences to ensure consistency on publication patterns and distribution of contributorships. For this, we used the Dutch NOWT Classification which introduces three levels of categorization: 7 broad areas, 14 fields, and 34 subjects. This classification is linked to the the Web of Science subject categories (see correspondence here https://www.cwts.nl/pdf/nowt_classification_sc.pdf). The classification is made at the journal level, which implies that, given the high incidence of the PloS One papers in our data set, most publications would be categorized as Multidisciplinary. To overcome this issue, publications in Multidisciplinary were reclassified into other more specific fields based on their reference lists. We identified the journal to which each of the references of the publications in our data set belong to. Then, we assigned to each publication the field from which most of its references come from. Finally, we only include those which are assigned to the Medical and Life Sciences fields. A total of 70,694 publications and 347,136 distinct authors were extracted from this process, constituting the ''seed data set''.

## Publication history of individual scientists

We reconstructed the publication histories of scientists, and predicted their contributions throughout their careers. The set of authors identified is retrieved from the seed dataset to ensure consistency on the predictions of the Bayesian Network model. But a series of thresholds are imposed. First, we retrieve authors' gender using the following sources to identify gender: Gender API, Genderize.io and Gender Guesser. We apply a 90% accuracy threshold before assigning gender and only include those authors who surpass such threshold. By promoting accuracy over recall, we assume some selection biases derived from limitations on the identification of gender. By doing so, we minimize potentially misclassified authors due to the assumption of gender as a binary variable. Second, we include only authors whose first publication occurred from 1980 onwards. While the CWTS in-house database includes publications prior to 1980, it does not contain metadata of sufficient quality as to rely on the name disambiguation algorithm. Hence, authors with their first publication prior to 1980 are discarded. Third, we include only authors who have contributed to at least five publications. We do this for two reasons. On the one hand, we remove transient authors, that is, those who have published sporadically, and focus only on scientists that have more chances of being pursuing a research career. On the other hand, this increases the accuracy of the author name disambiguation performed on those researchers. This is specially relevant since the algorithm adopts a conservative approach: when confronted with individuals having outlier patterns of behavior, such as rapid shifts across publication venues, disciplines and co-authors, it will consider them as different authors and consequently split their publications across different ''individuals''. Hence, by including a publication threshold, we focus on those individuals for whom the algorithm is more robust and accurate at identifying them uniquely. Last, we remove the publications classified as letters to ensure consistency between the two datasets with respect to the document type. As a result, the final dataset contains a total of 222,925 individuals and 6,236,239 distinct publications. The reason for the much larger set of publications is that for those scientists identified in the Seed dataset, we have expanded to all their other publications identified by the algorithm (and not just those from *Table 2*).

## Bayesian networks for predicting contributorships

Bayesian networks (BNs) graphically depict interactions among dependent multivariate data. The network structure represents a directed acyclic graph (DAG), where nodes represent random variables and arcs encode direct influences. Along with dependence statements, a BN encodes conditional independence statements among random variables. These conditional independencies are described by the d-separation concept (*Nielsen and Jensen, 2009*) and are captured graphically by the BN structure. The Markov property ensures a convenient factorization of the joint distribution of the multivariate data. Say $n$ continuously distributed random variables $X_1, X_2, \ldots, X_n$ are modeled by a Bayesian network. Then, the joint probability density function can factorize in the following manner

$$f(x_1, x_2, \ldots, x_n) = \prod_{i=1}^{n} f(x_i | P_a(X_i)), \qquad (1)$$

where $P_a(X_i)$, for $i = 1, \ldots, n$, represents the parent set of node $X_i$, that is, the set of nodes (variables) whose arcs are directed at $X_i$. The conditional densities $f(x_i | P_a(X_i))$, for $i = 1, \ldots, n$, of each random variable conditioned on its set of parent nodes encode the Markov property. The joint density factorization therefore depends on the structure of the network, that is, on the presence or absence of arcs and their directions.

There are numerous structures that can be considered, and the number of structures grows super-exponentially with the number of variables (*Robinson, 1977*). Let $a_n$ denote the number of BNs with $n$ random variables. Then

$$a_n = \sum_{k=1}^{n} (-1)^{k+1} \binom{n}{k} 2^{k(n-k)} a_{n-k}, \qquad (2)$$

where $a_0 = 1$. The structure of a BN can be learned from data or from experts, or from mixing data-driven algorithms with expert input.

Data driven learning algorithms of a BN structured are broadly categorized into constraint-based and score-based learning algorithms (*Scutari and Denis, 2014*). Constraint-based methods rely on conditional independence tests, whereas score-based methods employ likelihood-based metrics to evaluate structures. Both types of algorithms also contain a search procedure, such as a local search in the space of network structure (*Scutari and Denis, 2014; Koller and Friedman, 2009*). We employ the Max-Min Hill-Climbing (MMHC) algorithm

(*Tsamardinos et al., 2006*), which combined techniques from constraint and score-based algorithms, along with an initial local discovery algorithm of edges without any orientation.

We have employed a mixed approach, which imposed, via a white list, the direct influences of bibliometric to contributorship variables. The white-listed arcs are depicted in red in Figure S1. It is noteworthy that the arcs were present from employing the MMHC data-driven algorithm, and only the direction was switched. These white-listed nodes have been accounted for in learning the structure with the remaining variables. Thus, the remaining arcs in the BN together with their directionality have been fully assigned by using data-driven algorithms.

Finally, the BN structure has been subjected a robustness check by employing a bootstrap procedure, by which bootstrap replications of the data have been sampled 50 times from the initial data, with replacement. The bootstrap samples had the same size as the initial dataset. The MMHC algorithm has provided network structures and the arcs that have appeared in at least 80% of the structures have been retained. *Figure 2B* illustrates the resulting network.

### Cross-validation

To validate the BN used to predict contributions, we perform a k-fold cross-validation. The data are split in 10 subsets. For each subset, in turn, the BN is fitted on the other k - 1 subsets and a predictive loss function is then computed using that subset. Loss estimates for each of the k subsets are then combined to give an overall predictive loss. Since we are interested in predicting whether a scientist had a certain contributorship for the publications in the dataset, we translate the predictive loss into classification error. That is, we quantify the classification error rate of the BN in predicting a certain

contributorship, given the bibliometric information of scientists and publications. The classification error rates obtained for each contributorship with a cut-off value of 0.5 are shown in *Table 3*. While the error rates obtained are quite low, it is true that this validation is performed using data which is of the same nature as the data on which the BN has been quantified. This means that the extent to which contribution patterns in our dataset can be inferred to other datasets should be further investigated using different journals or fields.

### Constructing scientific profiles

#### Data aggregation

Predicted probabilities of all contributorship types obtained from the BN are available for each author-publication combination. We aim to aggregate those prediction at the author level, that is, to derive, for each scientist, the probability of fulfilling each contribution role. For this, we used the median probability value per contribution type. Furthermore, we grouped the publications by career stage, that is, publications within 5 years from the first publication (junior stage), publications between 5 and 15 years from first publication (early-career), publications between 15 and 30 years from first publication (mid-career) and publications after 30 years from first publication (late-career). Here must note that the selection of the time periods was selected for convenience and that any other division could have been selected. For each researcher, we obtain a median probability per contribution type and career stage.

Suppose within career stage $i$, with $i = 1, \ldots, 4$, a scientist has $k$ publications. Let $p_j^i$ the probability that the scientist performs contributorship $j$ within career stage $i$, for $j = 1, \ldots, 5$ denoting the five different types of contributions (WR, CT, CE, PE, AD). Then

**Table 3.** Classification error rates from cross-validation of Bayesian Network model for the contribution variables.

For contributorships, the percentage of mis-classified predictions is shown, while for NC, the mean squared error between the predicted and the observed values is reported.

| Variables | Min. | Median | Mean | Max. |
|---|---|---|---|---|
| WR | 0.062 | 0.064 | 0.064 | 0.065 |
| AD | 0.064 | 0.067 | 0.067 | 0.069 |
| PE | 0.072 | 0.075 | 0.075 | 0.077 |
| CE | 0.062 | 0.064 | 0.064 | 0.066 |
| CT | 0.077 | 0.078 | 0.078 | 0.081 |
| NC | 0.120 | 0.125 | 0.125 | 0.127 |

$$p_j^i = Median(p_{j,1}^i, p_{j,2}^i, \ldots, p_{j,k}^i), \qquad (3)$$

where $p_{j,1}^i$ is the predicted probability for contribution $j$ of scientist's first publication in career stage $i$. For the number of contributions (NC), the same aggregation rule is applied

$$NC^i = Median(NC_1^i, \ldots, NC_k^i) \qquad (4)$$

where $NC_1^i$ is the predicted number of contributions for the first paper in career stage i.

## Robust archetypal analysis

Profiles of researchers, by career stage, are obtained using a robust archetypal analysis. Archetypal analysis aims to identify archetypes that emerge from the given contribution data for scientists. This approach has been previously applied to identify scientists' profiles based on citation and publication data (*Seiler and Wohlrabe, 2013*). The archetypes are extreme observations in a multivariate dataset and represented as convex combinations of the observations in the dataset that result from a least squares problem (*Cutler and Breiman, 1994*). For multivariate data with n observations (scientists, per career stage, in our case) and m random variables (types of contributorships, in our case), then X is a n×m matrix denoting the aggregated dataset. For given k archetypes, denote by Z the k×m the matrix of archetypes, represented in terms of the types of contributorships. Then, the residual sum of squares (RSS) plotted in Figure S1 is denoted by

$$RSS = ||X - \alpha Z^T||_2, \qquad (5)$$

with $Z = X^T\beta$, where $\alpha, \beta$ are positive coefficients and where $||\cdot||_2$ denotes the Euclidean matrix norm. In turn, each observation in the dataset can be represented as a convex combination of the archetypes

$$X \approx \alpha Z^T. \qquad (6)$$

In the standard approach of archetypal analysis, each residual contributes to the RSS with equal weight. The archetypal analysis is thus sensitive to outliers, whose large residuals can contribute significantly to the RSS. A robust archetypal analysis (*Eugster and Leisch, 2011*) has been proposed to weight down the influence of outliers to the construction of archetypes. By letting W be a n×n matrix of weights, we define the weighted RSS

$$RSS = ||W(X - \alpha Z^T)||_2. \qquad (7)$$

The weights can be chosen by the user or can be chosen to depend on each observation's residual. The robust archetypal analysis proposed by *Eugster and Leisch, 2011* proposes an iterative re-weighted least squares algorithm. Unlike the k-means clustering approach, which engages averaging when profiling out clusters, archetypal analysis focuses on extremes and explore the heterogeneity of complex multivariate data. Furthermore, archetypes are not forced to be mutually exclusive, as principal components are, nor do they remain the same when the number of considered archetypes is changing. The archetypal analysis has been performed using the archetypes package in R (*Eugster and Leisch, 2009*).

## Confidence intervals, hypothesis testing and effect sizes

In this section we report how confidence intervals were constructed for *Figures 4*, *6* and *7*, as well as additional analyses conducted for those figures which are reported in supplement figures.

Neither the BN's predictions, nor the RAA coefficient or alpha values account for uncertainty in the form of confidence intervals. Nevertheless, accounting for uncertainty in reporting the coefficient values of the archetypes would inform about the potential varying influence of contributions within each archetype. To construct those uncertainty intervals for the coefficient values of contributions (*Figure 4*), we have used the classification error rates as uncertainty bounds of our predicted probability of contribution. We have extracted and added the mean classification error rate to our predicted values, creating two additional datasets of predicted probabilities of contributorships: one for the upper value and the other one for the lower value. We have then conducted a RAA for each of the two new datasets, for each career stage. The same number of archetypes have been obtained as for the initial dataset. The coefficient values obtained for the upper and lower datasets are reported as uncertainty intervals in *Figure 4*.

When analyzing the distributions of number of publications and share of highly cited papers by archetype across career stages (*Figure 5*), we tested if the differences in medians are statistically significant. We performed a Wilcoxon rank sum hypothesis test. Very low p-values ($p - value < 2.2E - 16$ for all paired hypothesis tests), suggest that the differences in median are statistically significant within each career stage.

However, this was expected, given the large sample sizes across the career stages. To evaluate the strength of the differences, we also investigated the effect size using the epsilon squared measure (*Kelley, 1935*), which are displayed, along with confidence intervals, in *Figure 5—figure supplement 1*.

In the case of gender and author position (*Figures 6* and *7*), we investigated whether the differences in proportions within each career stage are statistically significant by using the two-proportion z-test (*Cohen, 2013*). The Pearson's chi-squared test statistics allowed us to determine 95% confidence intervals for the differences in proportions, which, in turn, has been used to construct 95% confidence intervals for each proportion. Also, we reported effect sizes using *Cohen, 2013*.

### *Limitations of the study*
#### Representativeness of the sample of scientists
The analysis is based on a set of publications and a sample of scientists which may not represent accurately the whole population of scientists. This means that, despite the robustness of the results, any inference to the whole population should be done with caution. Furthermore, the thresholds imposed to introduce such scientists in the archetypal analysis further restricts such inference endeavour. If we compare the productivity distributions of our set of researchers and for the whole population of the Web of Science, we observe that while we still retain a high skewness of productivity, this is much lower than the overall one.

#### Identification of scientists
The study relies heavily on the competence of an author name disambiguation algorithm to correctly identify disambiguated authors. As previously noted, this algorithm has some limitations which are partially overcome by the production thresholds imposed. However, inaccurate assignments can still occur.

#### Author age
We estimate researchers' age based on the year of first publication and build the four career stages based on such year. However, alternative approaches could have been adopted and these could have some impact on the results. For instance, first year of first-authored publication could have been used instead. The selection of the first year of publication is based on empirical

data suggesting that it is the best proxy for PhD year (*Nane et al., 2017*).

## Taxonomy of contributorships
In this paper, contributions are classified into five types. These types are obtained from the data itself. However, one may question the appropriateness of the number and contribution types. The ones used in this paper are consistent with those used in other studies (*Larivière et al., 2016*), but different from those implemented in the CRediT initiative, which defines up to 14 types of contributions. Furthermore, evidence suggests that author self-reporting on contributorship is not exempt of limitations (*Ilakovac et al., 2007*). Questions like the extent to which contribution types are field-dependent are still unsolved. With this respect, our predictions already point towards some of these issues. Despite the low error rates, we observe that the distribution of predicted probabilities exhibits a normal distribution for writing the manuscript (*Figure 2A*). This could be due to the ambiguity of the statement. As observed in the CRediT intitiative, this statement is disclosed into two: wrote the first draft and wrote parts of the manuscript and revised. Such distinction might help the model to better discriminate contributorships.

## Acknowledgements
The authors thank Lidia Carballo-Costa and Alejandro Quintela del Río (Universidade da Coruña, Spain) for their support in applying the archetypal analysis. This work was supported by the European Union's Horizon 2020 research and innovation programme under the Marie Skłodowska-Curie grant agreement No 707404 and the South African DST-NRF Centre of Excellence in Scientometrics and Science, Technology and Innovation Policy (SciSTIP). The authors declare that they have no competing interests. Data are derived from Clarivate Analytics Web of Science. Copyright Clarivate Analytics 2020. All rights reserved.

**Nicolas Robinson-Garcia** is in the Delft Institute of Applied Mathematics, Delft University of Technology, Delft, Netherlands
elrobinster@gmail.com
https://orcid.org/0000-0002-0585-7359
**Rodrigo Costas** is at the Centre for Science and Technology Studies (CWTS), Leiden University, Leiden, The Netherlands, and the Centre for Research on Evaluation, Science and Technology (CREST), Stellenbosch University, Stellenbosch, South Africa

https://orcid.org/0000-0002-7465-6462

**Cassidy R Sugimoto** is in the School of Informatics, Computing, and Engineering, Indiana University Bloomington, Bloomington, United States

**Vincent Larivière** is in the École de bibliothéconomie et des sciences de l'information, Université de Montréal, Montreal, Canada

**Gabriela F Nane** is in the Delft Institute of Applied Mathematics, Delft University of Technology, Delft, Netherlands

*Author contributions:* Nicolas Robinson-Garcia, Conceptualization, Data curation, Software, Formal analysis, Funding acquisition, Visualization, Methodology, Writing - original draft; Rodrigo Costas, Formal analysis, Supervision, Validation, Methodology, Writing - review and editing; Cassidy R Sugimoto, Formal analysis, Supervision, Validation, Writing - review and editing; Vincent Larivière, Resources, Formal analysis, Supervision, Validation, Writing - review and editing; Gabriela F Nane, Conceptualization, Formal analysis, Supervision, Funding acquisition, Validation, Methodology, Writing - original draft, Writing - review and editing

*Competing interests:* Cassidy R Sugimoto: Reviewing Editor, eLife. The other authors declare that no competing interests exist.

**Funding**

| Funder | Grant reference number | Author |
| --- | --- | --- |
| European Commission | 707404 | Nicolas Robinson-Garcia |
| South African DST-NRF Centre of Excellence in Scientometrics and Science, Technology and Innovation Policy | | Rodrigo Costas |

The funders had no role in study design, data collection and interpretation, or the decision to submit the work for publication.

**Decision letter and Author response**
Decision letter https://doi.org/10.7554/eLife.60586.sa1
Author response https://doi.org/10.7554/eLife.60586.sa2

## Additional files

### Supplementary files

- Transparent reporting form

**Data availability**

All data is openly accessible at http://doi.org/10.5281/zenodo.3891055.

The following dataset was generated:

| Author(s) | Year | Dataset URL | Database and Identifier |
| --- | --- | --- | --- |
| Robinson-Garcia N, Costas R, Sugimoto CR, Larivière V, Nane GF | 2020 | https://zenodo.org/record/3891055 | Zenodo, 10.5281/zenodo.3891055 |

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

## Appendix 1

## Future directions on profiling diversity in research careers

### Stability of profiles based on contribution taxonomies

In this paper we profile scientists based on an specific taxonomy of contributorships with five different statements. However, other taxonomies including the more comprehensive proposed by CReDiT *Allen et al., 2014* have also been suggested and are implemented in journals. The extent to which the archetypes identified in this study are sensitive to such taxonomies remains to be explored. Furthermore, some of the original contributorships were removed from the final analyses due to the lack of data on authors reporting them. These two contributorships were 'Other' and 'Approved/Revised final version of the manuscript'. Hence one could question which would be the more appropriate contributorship statement, as well as the appropriate number in order to be meaningful for most publications and at the same time sufficiently detailed as to provide rich information on different types of contributorship.

### Value of contribution statements

Related with the previous question, one might wonder how to disentangle contributorship when more than one author state to have contributed in the same tasks. Is their contributorship for each of the common tasks equal? Is author A more involved in some tasks than author B? How could this be reported in a consistent and quantifiable way? Some of these issues have already been treated with regard to author order, with different types of author counting proposed in case of multi-authored papers (*Waltman, 2016*). How this could be resolved or whether this is an issue that should be addressed should be better explored in the future. Considering that the inclusion of contribution statements is derived from an effort to provide more transparency and go beyond the limitations of authorship especially in multi-authored publications, can the fact that this is self-reported information be misleading in some cases? Furthermore, if this type of statements are to be used and scrutinized in an evaluative context which is highly competitive, it could lead to further disputes and misbehaviour related to contribution disagreements (*Smith et al., 2020*).

### Reporting uncertainty

From a methodological standpoint, the analyses conducted in this study presented certain challenges when reporting uncertainty. This is due to the fact that neither methods provide confidence intervals to their estimates. The BNs predictions probabilities already account for uncertainty, and RAA does not report any uncertainty. We therefore opted for using the error rates resulted from the cross validation analysis to obtain confidence intervals around the coefficient values of contributions by archetype. We also used the confidence intervals of the employed test statistics when testing for statistically significant differences in median publications and share of highly cited publications and proportion of researchers distributed with respect to gender and author position. However, other methods can be employed, such as probabilistic archetypal analysis (*Seth and Eugster, 2016*), which we believe is worth exploring further.

### Longitudinal analyses of archetypes

In our study we distinguish between four different stages of a research trajectory, namely junior, early-, mid- and late-career. These stages go in consistency with alternative proposals in the literature (*Laudel and Gläser, 2008*; *Milojević et al., 2018*; *European Commission, 2016*). However, one could question if shorter periods of time could be analyzed to explore the stability of the profiles identified. For instance, a yearly analysis could be proposed to identify shifts of profiles over time, but the reliance on productivity may hamper the robustness of the findings or would be only limited to highly productive scientists.

Career trajectories, prediction and causality

Finally, we focus on career trajectories and on factors which influence the length of scientists' trajectories. An appropriate analysis of researchers' career trajectories requires an appropriate dataset. Our current dataset, albeit extensive, includes only researchers who published in PloS journals and only over a period of seven years. Extending the analysis over a larger period would include different cohort of scientist with different trajectory lengths and would hence allow for meaningful insights on trajectories, as well as a robustness analysis of the archetypes. We note that, with this respect, our current dataset is somewhat limited, as our most senior scientists have only had a long career up until 2006–2013. Furthermore, a dataset covering more journals would allow to test the methods further and available contribution data would enable the evaluation of the BN's predictive performance.

Our current analysis pointed out association between archetype assignment at current stage with assignment at previous career stage, or with gender. A prediction model would allow us to find the best predictor or best combination of predicting factors for career advancement. With this respect, a dynamic BN would account for the temporal dependencies. Once again, a more extensive dataset would enable this analysis. Furthermore, properly assessing the influence of various factors would invoke causal inference. An overview of the statistical method is provided in *Pearl, 2009*. Counterfactual analysis appears to be germane. Additionally, the transition between archetypes within career advancement can be accounted for in the g-computation (*Yu and van der Laan, 2002*).

