## [Decision Letter]

Thank you for submitting your article "Task specialization and its effects on research careers" to *eLife* for consideration as a Feature Article. Your article has been reviewed by two peer reviewers, and the evaluation has been overseen by the *eLife* Features Editor (Peter Rodgers). The following individuals involved in review of your submission have agreed to reveal their identity: Allison Morgan (Reviewer #1).

The reviewers and editors have discussed the reviews and we have drafted this decision letter to help you prepare a revised submission. We hope you will be able to submit the revised version within two months.

Summary:

Using data on over 6M publications and 200K authors, the authors aim to consider how prominent various scientific archetypes are across researchers' careers and whether there are gendered differences in these roles. The main takeaways are: (i) contributions shift over career stages, due to who is performing experiments/analyzing data (junior/early-career) or conceiving experiments/writing (mid/late-career); (ii) researchers who present as leaders in their early-careers are more likely to remain; (iii) women are less likely to appear in the leader archetype across career stages (and more likely to be in specialized/supporting roles with typically lower productivity). However, there are a number of points and concerns about the work that need to be addressed to make the article suitable for publication.

Essential revisions:

1) Title: I worry that the phrase "its effects" implies a causal relationship that is not warranted by the analysis presented in the paper. Possibly something like: "Task specialization across research careers" is a little more accurate.

2) Please explain why Bayesian analysis was used to determine which of the variables in the dataset predict the Contributorship categories: to me the use of machine learning with boosting classification would seem more appropriate.

Also, please explain how various of the criteria used in the analysis were chosen (eg, the definitions of career stage, the number of iterations used in the Bayesian analysis, the ICMJE criteria included in the analysis).

3) I would like to see the analysis of Contributorship and career stages across years, if possible, to see if there is a change in trends over the years, especially for the gender differences and career advancement.

[Note from the Features Editor: Addressing this point is optional]

4) The low error rates in the Bayesian analysis seem very impressive, but to interpret them better I'd appreciate some notion of recall since (I think?) most authors would not fill a given role. I also assume a cutoff value was chosen from the predicted probability, but this was not explicitly said. These suggestions would likely not impact the conclusions of the work, but would aid in reproducibility.

5) I do not understand why authors use Spearman correlation coefficient because you were investigating an association between nominal/categorical variables and ordinal variables. Please use appropriate measures of association for variables of interest or omit this analysis completely.

6) In subsection “Bayesian network model for predicting contributorship”: The strength of the arcs, i.e., relationships between variables, has been investigated using the bootstrap procedure, with 50 repetitions. Only the arcs that were present in 80% of the repetitions have been considered and are depicted in Figure 2B.

Is this standard procedure? If it is, please give a reference, and in case it is not please explain what is the rationale behind choosing those criteria.

There are several places where I believe the authors should provide statistical hypothesis testing and state the actual differences. These are documented below:

7) In subsection “Career paths, productivity and citation impact”: Figure 5B/C doesn't lead me to believe the differences in publications or citations are meaningfully different across career stages, but the text alludes to differences. Mentioning whether these are supported by statistical tests would strengthen your claims. Also, please expand the caption for figure 5 to fully explain what is being plotted in panels B and C.

8) Figure 6: A significance test across the proportions for men and women would be valuable again, since the text draws our attention to them.

9) Figures 6 and 7: Please include confidence intervals so that the reader can assess whether there are differences between groups.

10) You argue that there are differences in proportions between men and women in archetypes, but I wonder whether that is really due to the fact because in the previous paragraph you state that is more likely for researchers who are leaders in early stages to come to the late stages of career. I wonder what the more predictive factor for career advancement is - gender or specialization in early stage - and it would be good if the authors could resolve this issue by using some prediction model.

Also, you state that the shares of leader archetype are consistently lower for women compare to the men. Can you please test this using some kind of test of proportions (chi square for example) and also report the 95% confidence intervals for proportions?

Also, what does the distribution of \alpha (weights on archetypes) look like for most researchers? Is it easy to distinguish which archetype researchers fall into at a given career stage? This seems slightly important to the interpretation of the results.

---

## [Author Response]

[We repeat the reviewers’ points here in italic, and include our replies point by point, as well as a description of the changes made, in Roman.]

We thank the reviewers for their thoughtful reading and constructive comments and suggestions to our earlier manuscript. We have carefully edited the manuscript and have attempted to address all reviewer concerns. The manuscript has improved as a result. Following we describe the major changes:

# we have described, where possible, the uncertainty around our estimates;

# we have included results of hypothesis tests which were indicated by the reviewer, along with effect sizes; and

# we have included an appendix section with future directions on profiling diversity in research careers, which addresses some of the reviewer’s comments.

Essential revisions:1) Title: I worry that the phrase "its effects" implies a causal relationship that is not warranted by the analysis presented in the paper. Possibly something like: "Task specialization across research careers" is a little more accurate.

The reviewer is correct: the title indeed implied causality, which is not supported by our analysis. We have modified the title following the reviewer’s comment. We have also added an Appendix in which we discuss potential ways in which causality could be studied as further research.

2) Please explain why Bayesian analysis was used to determine which of the variables in the dataset predict the Contributorship categories: to me the use of machine learning with boosting classification would seem more appropriate.

We have employed Bayesian Networks (BNs) to predict types of contributions, by using the existing dependence relationships within our multivariate dataset. We did not use the BN specifically to select among the predictor variables. However, the resulting network, which was obtained using a best-fitting criterion, as well as the bootstrap step, confirmed the joint influence of the predictor variables on the contributions. We emphasize that, unlike in the regression setting, where one can identify partial influence of each predictor, BNs model the joint influence of the predictor variables. We have now added the following sentences at the beginning of the “Bayesian network model for predicting contributorship” subsection:

“BN accounts not only for dependencies between the predictor variables and variables of interest, but also for dependencies between predictor variables. This characteristic, along with the forthright graphical representation makes BNs an attractive choice to model dependent multivariate data”.

We thank the reviewer for suggesting the boosting classification. To the best of our understanding, boosting encompasses algorithms which train a sequence of weak models to improve classification performance. The models can even be used with BNs (see, for example, Jing et al., 2008) to compensate for the goodness-of-fit focus of the learning algorithms with the prediction accuracy. Nonetheless, we have shown that our BN provides accurate classification in the cross-validation analysis. Furthermore, rather than predicting class labels, we were interested in obtaining prediction probabilities, which have been later used in our archetypal analysis.

We have considered 50 bootstrap samples to investigate the robustness of the BN structure obtained. The choice of 50 bootstrap samples is partly due to the computational burden this procedure implies. We have also added at the end of the second paragraph of the “Methods and Materials” section two sentences indicating the machine on which the analyses were conducted and the approximate required computational time. We mention that learning the BN’s structure and the bootstrap procedure were the most time intensive parts of our analyses.

Also, please explain how various of the criteria used in the analysis were chosen (eg, the definitions of career stage, the number of iterations used in the Bayesian analysis, the ICMJE criteria included in the analysis).

The four career stages are defined *ad hoc* but are consistent with what we have observed in other research careers studies or classifications. For instance, the European Commission distinguishes between *first stage researcher* (up to PhD), *recognised researcher* (PhD-holders still not fully independent), *established researcher* (independent researchers) and *leading researcher* (those with a leading role in their field) (European Commission, 2016). Laudel and Gläser refer to the *apprentice* (learning phase), *colleague* (independent researcher), *master* (independent researcher and mentor), and *elite* (leading researcher in their field) (Laudel and Gläser, 2008). These classifications go, to some extent, in accordance with the one we propose, but already introduce some judgment on the role researchers play at each stage (which we try to surface with our analyses). Nonetheless, no classification provides any temporary indication to operationalise the career stages.

Another proposal, also similar to our four stages is that by Milojevic et al (Milojević et al., 2018). In this case, they do propose temporary boundaries between stages. We employ their terminology, with a slight change in the time periods. Specifically, the full career stage is chosen as >20 years after first publication. We choose the threshold of 30 years, to better identify seniority in our analysis. To clarify this, we have added the following sentence at the end of the first paragraph in the “Results” section:

“These four stages are defined in consistency with other classifications of career stages in the literature (Laudel and Gläser, 2008; Milojević et al., 2018; European Commission, 2016)”

With regard to the ICMJE criteria, we do not use it in our analysis, but refer to it (as to other studies) to mention and test (Figures 1B and 7) whether the common assumption on relating contributorship or task role with author position is observed in our data. In the revised version of the manuscript we have highlighted that this is a common assumption. For this, see the first paragraph of the section “Results”.

3) I would like to see the analysis of Contributorship and career stages across years, if possible, to see if there is a change in trends over the years, especially for the gender differences and career advancement.[Note from the Features Editor: Addressing this point is optional.]

We thank the reviewer for the suggestion. Indeed, it would be interesting to see the archetypal analysis per year. Our only concern (apart from the computational constraints) is that there might be high variations in the number of publications of researchers per year. For example, one researcher might have 10 publications in year 1, no publications in year 2, and 2 publications in year 3. Constructing archetypes and, more importantly, assigning researchers to archetypes based on a few number of publications is not desirable, as the archetypal analysis might not be so robust. Moreover, the archetypes will be sensitive to outliers and noise in general and we will probably not see the 2-3 archetypes so clearly.

To have an indication of the abovementioned variability, we have plotted the number of publications each scientist produces every year, by career stage (see Author response 1). As you can see, while the productivity distribution is highly skewed, still, the majority of scientists publish below 5 publications a year.

In any case, we believe that the point raised by the reviewer is of great interest and should be further explored. We have addressed the issue of stability of archetypes and the ideal time frame in the new Appendix 1.

4) The low error rates in the Bayesian analysis seem very impressive, but to interpret them better I'd appreciate some notion of recall since (I think?) most authors would not fill a given role. I also assume a cutoff value was chosen from the predicted probability, but this was not explicitly said. These suggestions would likely not impact the conclusions of the work, but would aid in reproducibility.

We thank the reviewer for this comment. Since we were interested not only in accurately predicting the presence of a contribution, but also when the contribution was not present, our error rate accounted for both type I and type II errors, cumulating both false positive and negative. Nonetheless, we present in Author response table 1 the recall and precision computed for our cross-validation analysis.

**Author response table 1. resptable1:** 

	WR	AD	CE	CT	PE	
Precision	0.98	0.88	0.89	0.89	0.87	
Recall	0.89	0.99	0.99	0.99	0.99	

We used 0.5 as the cut off value. This is now mentioned in the subsection “Crossvalidation”, when referring to Table 3.

Reviewer’s comment that “*most authors would not fill a given role*” stirred our curiosity, so we investigated the likelihood that authors fill a certain contribution role. For our initial data, the percentage of authors performing a certain contribution is given in the ‘mean’ column of Author response table 2 below. The lowest percentage is for CT (contributed with tools), with 35% of the authors performing this task. The results suggest that the data is quite balanced. Finally, we mention that the archetypal analysis showed that all contributions have been identified by the predictions of BN.

**Author response table 2. resptable2:** 

WR	517008	0.48	0.50
AD	517008	0.52	0.50
CE	517008	0.48	0.50
CT	517008	0.35	0.48
PE	517008	0.51	0.50
NC	517008	2.46	1.32

5) I do not understand why authors use Spearman correlation coefficient because you were investigating an association between nominal/categorical variables and ordinal variables. Please use appropriate measures of association for variables of interest or omit this analysis completely.

We thank the reviewer for pointing this out. The ‘cor’ function in R uses adapted correlation measures for discrete variables. However, we could not find a proper reference and decided to use the functions in the ‘psych’ package in R, which documents the correlation coefficients for binary RVs, as well as for mixed RVs (between continuous and discrete RVs). Tetrachorics correlation is used for binary RVs and polyserial or biserial is used for the mixed RVs. We have therefore obtained a mixed correlation matrix, which is slightly different from the previous correlation matrix in which Spearman correlation is only used for continuous variables. We note that the differences in coefficients are not that big though. This means that Figure 2A is now slightly modified. Also we have modified the text as follows:

References to Figure 2A now reflect new values in the figure.

“Methods and Material” section, last sentence in the first paragraph. The following sentences were added:

“Mixed correlation matrix in Figure 2A was calculated using the psych package (Revelle et al., 2010). Spearman rank correlations have been determined for continuous variables, tetrachoric correlations for binary random variables and biserial and polyserial correlations for mixed random variables, i.e., between binary or other discrete and continuous random variables.”

6) In Results subsection “Bayesian network model for predicting contributorship”: The strength of the arcs, i.e., relationships between variables, has been investigated using the bootstrap procedure, with 50 repetitions. Only the arcs that were present in 80% of the repetitions have been considered and are depicted in Figure 2B.Is this standard procedure? If it is, please give a reference, and in case it is not please explain what is the rationale behind choosing those criteria.

There is no standard procedure. Since the bootstrap procedure introduces some variability, we chose 80% as a robust check against the variability. We did try additional thresholds at 95%, 90% and 75% to check for robustness and did not observe major differences on the final BN structure. Regarding the number of repetitions, there is no standard procedure either. We opted for 50 repetitions to have a balance between computational time and robustness. The bootstrapping procedure to develop the BN took around 20 hours.

There are several places where I believe the authors should provide statistical hypothesis testing and state the actual differences. These are documented below:7) In subsection “Career paths, productivity and citation impact”: Figure 5B/C doesn't lead me to believe the differences in publications or citations are meaningfully different across career stages, but the text alludes to differences. Mentioning whether these are supported by statistical tests would strengthen your claims. Also, please expand the caption for figure 5 to fully explain what is being plotted in panels B and C.

We thank the reviewer for suggesting hypothesis testing. We point out that employing a hypothesis test does not answer the question whether the differences are meaningful, but if the differences have occurred by chance or not, assuming the null hypothesis that there is no difference. Nonetheless, we think that hypothesis testing is adding value to our analysis. For productivity and citation impact, we have performed a Wilcoxon rank sum hypothesis test to investigate whether the differences in median publications and median share of highly cited publications for archetypes differ statistically significantly within each career stage.

As expected, due to the large sample sizes across the career stages, we obtained very low p-values (p-value < 2.2e-16 for all tests). To evaluate the strength of the differences, we also investigated the effect sizes and reported the results in the paper, in the sections Results and Methods and Materials. We found large effect sizes for differences in number of publications for early and mid career, and medium for late career. For share of highly cited papers, the effect size was found to be small for all career stages. The results are includes in Figure 5 - Figure supplement 1.

Finally, the caption of figure 5 has been expanded giving full detail of the indicators depicted in panels B and C.

8) Figure 6: A significance test across the proportions for men and women would be valuable again, since the text draws our attention to them.

We have employed a two-proportion z-test, which tests that the proportions in two groups are the same. Pearson’s chi-squared test statistics allowed us to determine 95% confidence intervals for the differences in proportions, which, in turn, has been used to construct 95% confidence intervals for each proportion. The confidence intervals have been graphically included in Figure 6. Once again, the very small p-values, as well as the confidence intervals indicate statistically significant differences. However, the power analysis revealed a small and medium effect size, which is included in Figure 6-Figure supplement 3.

9) Figures 6 and 7: Please include confidence intervals so that the reader can assess whether there are differences between groups.

We have included confidence intervals, as mentioned at the previous point. With respect to the reviewer's comment about “whether there are differences between the groups”, we point out that hypothesis testing and confidence intervals are equally informative on whether differences between groups are *statistically* significant.

10) You argue that there are differences in proportions between men and women in archetypes, but I wonder whether that is really due to the fact because in the previous paragraph you state that is more likely for researchers who are leaders in early stages to come to the late stages of career. I wonder what the more predictive factor for career advancement is - gender or specialization in early stage - and it would be good if the authors could resolve this issue by using some prediction model.

This comment raises a very interesting point, that is, which are the predictive factors for career advancement, gender or archetype, or other(s)? Nevertheless, we have agreed that a proper predictive analysis should be based on an appropriate dataset. But it is certainly a direction we would like to consider in the future. We have mentioned this in Appendix 1, career trajectories, prediction and causality. For the current manuscript, we have introduced two supplemental figures for Figure 6, in which we illustrate Sankey diagrams by gender. These figures suggest that it is the difference in the distribution of scientists by gender at the early stages which seem to affect later differences, but it also seems that career advancement is affected by scientists’ archetype.

Of course, these statements need to be supported to proper statistical analysis. For now, they are only suppositions, and are presented as such in the revised version of the manuscript. First, we have removed the sentence “As observed, a gender disparity on the distribution by archetype and stage is consistent in all career stages” from the “Archetypes and gender” subsection and this has been replaced with the following sentence:

“We observe a gender disparity especially in the early- and mid- career stages.”

We have also added the following sentence at the end of the subsection “Archetypes and gender”:

“These differences on the distribution of scientists by archetype and gender and how they might affect their trajectory is made more evident in figures S6-1 and S6-2.”

The Sankey diagrams also suggest a potential confounding effect of gender. That is, it seems that it is more likely to be in a leader position once having achieved the leader position in the previous career stage. However, the leader career stage is correlated with gender. Nonetheless, a proper predictive analysis would shed more light on this aspect, but, as mentioned earlier, would require an appropriate dataset to analyze.

With this respect, the (conditional) probabilities, based on the observed frequencies, presented in Author response table 3 below can support the claims above.

**Author response table 3. resptable3:** 

Probability	All	Women	Men
P(early Leader)	0.371	0.268	0.4314
P(early Leader| Junior Specialized)	0.4214	0.3379	0.2991
P(mid Leader)	0.356	0.249	0.415
P(mid Leader|early Leader)	0.685	0.606	0.71
P(mid Leader|early Specialized)	0.197	0.144	0.335
P(late Leader)	0.088	0.054	0.107
P(late Leader|mid Leader)	0.246	0.218	0.256

Also, you state that the shares of leader archetype are consistently lower for women compare to the men. Can you please test this using some kind of test of proportions (chi square for example) and also report the 95% confidence intervals for proportions?

We have used the two-proportion z-test to check if this difference has occurred by chance. The very small p-values indicate that it is very unlikely that the lower share of leader archetype for women to have occurred by chance, when we assume there is no difference (the null hypothesis). Moreover, specifically for the leader archetype, we obtain a medium size effect for early and mid career, the largest size effect within the gender analysis.

Also, what does the distribution of \alpha (weights on archetypes) look like for most researchers? Is it easy to distinguish which archetype researchers fall into at a given career stage? This seems slightly important to the interpretation of the results.

In order to look at differences between archetype assignments we compute the difference between the alpha scores for each pair of archetype and by stage. Alpha scores range from 0 to 1. As observed in the table below, median differences are relatively large (above 0.1) except for leader vs. specialized in the early-career stage and specialized vs. supporting at the mid-career stage. Still, in the former case the mean difference is 0.08. In the case of the former, the median difference is 0.06.

**Author response table 4. resptable4:** 

Junior stage	min	1st quartile	median	mean	3rd quartile	max
Spe. vs. supp.	-1,00	-0,87	-0,61	-0,50	-0,26	1,00
**Early-career**						
Lead. vs.spe.	-1,00	-0,25	0,08	0,08	0,44	1,00
Spe. vs. supp.	-1,00	-0,08	0,25	0,19	0,48	1,00
Lead. vs. supp.	-1,00	-0,13	0,15	0,10	0,39	1,00
**Mid-career**						
Lead. vs.spe.	-1,00	0,18	0,51	0,41	0,76	1,00
Lead. vs. supp.	-1,00	0,16	0,42	0,39	0,68	1,00
Spe. vs. supp.	-1,00	-0,14	0,06	0,02	0,23	1,00
**Late-career**						
Lead. vs. supp.	-1,00	-0,41	-0,17	-0,12	0,11	1,00

References:

Jing, Y., Pavlović, V., & Rehg, J. M. (2008). Boosted Bayesian network classifiers. *Machine Learning*, *73*(2), 155-184.

European Commission (2016): https://euraxess.ec.europa.eu/europe/career-development/training-researchers/research-profiles-descriptors

Laudel, G., & Gläser, J. (2008). From apprentice to colleague: The metamorphosis of early career researchers. Higher education, 55(3), 387-406.

Milojević, S., Radicchi, F., & Walsh, J. P. (2018). Changing demographics of scientific careers: The rise of the temporary workforce. PNAS. 115(50), 12616-12623.